

# Profile of the bile acid FXR-FGF15 pathway in the glucolipid metabolism disorder of diabetic mice suffering from chronic stress

Weijia Cai[1,2], Canye Li[1,2], Zuanjun Su[1,2], Jinming Cao[1,2], Zhicong Chen[1,2], Yitian Chen[1,2], Zhijun Guo[3], Jian Cai[4] and Feng Xu[1,5]

[1] Fengxian Hospital, Southern Medical University, Shanghai, China
[2] School of Pharmaceutical Science, Southern Medical University, Guangzhou, China
[3] Heyou Meihe Hospital, Foshan, Guangdong, China
[4] Fengxian Mental Health Center, Shanghai, China
[5] Sixth People's Hospital South Campus, Shanghai Jiaotong University, Shanghai, China

Corresponding authors
Jian Cai, 455320109@qq.com
Feng Xu, xuf@smu.edu.cn

## ABSTRACT

**Background**. Imbalances in bile acid (BA) synthesis and metabolism are involved in the onset of diabetes and depression in humans and rodents. However, the role of BAs and the farnesoid X receptor (FXR)/fibroblast growth factor (FGF) 15 signaling pathway in the development of diabetes and depression is still largely unknown. Therefore, we investigated the potential molecular mechanisms of BAs that may be associated with glucolipid metabolism disorders in diabetic mice subjected to chronic stress.
**Methods**. The type 2 diabetes mellitus (T2DM) mouse model was induced by feeding mice a high-fat diet and administering an intraperitoneal injection of streptozotocin (STZ). The chronic unpredictable mild stress (CUMS) procedure was performed by introducing a series of mild stressors. Forty mice were randomly divided into the regular chow feeding group and the high-fat diet feeding group. After two weeks of feeding, the mice were randomly divided into four groups: the Control group, CUMS group, T2DM group, and T2DM+CUMS group. The T2DM group and T2DM+CUMS group received an intraperitoneal injection of STZ to induce the T2DM model. The CUMS and T2DM+CUMS groups were exposed to CUMS to induce depressive-like phenotypes. Blood and tissue samples were obtained for pertinent analysis and detection.
**Results**. Compared with the T2DM mice, T2DM+CUMS mice had higher blood glucose and lipid levels, insulin resistance, inflammation of the liver and pancreas, impaired liver function, and increased total bile acids. These changes were accompanied by attenuated FXR signaling. Chronic stress was found to attenuate FXR expression and its downstream target, FGF15, in the ileum when compared with the T2DM group.
**Conclusion**. FXR may play a role in the diabetic disorder of glucolipid metabolism when aggravated by chronic stress. FXR and its downstream target, FGF15, may be therapeutic targets for treating comorbid T2DM and depression.

## INTRODUCTION

The International Diabetes Federation (IDF) estimated that 537 million people (10.5%) were living with diabetes in 2021 across the globe. This number is expected to rise to 783 million people (12.2%) by the year 2045 (*Sun et al., 2022*). Of these, 30–40% of patients with diabetes are expected to have at least one complication within 10 years of diagnosis, which may include macrovascular and microvascular diseases (*Viigimaa et al., 2020*). Diabetics are living longer lives and new complications are emerging, including cancer and infections, as well as negative impacts on psychological and mental health (*Harding et al., 2019*). Diabetic patients have an increased risk of developing major depressive disorder (*Ali et al., 2006*), anxiety (*Fisher et al., 2008*), or a severe mental illness such as schizophrenia (*Vancampfort et al., 2016*). Depression remains a significant risk factor in people with type 1 and type 2 diabetes *versus* those with normal blood glucose levels (*Semenkovich et al., 2015*). A systematic review and meta-analysis concluded that depression may develop in one out of five type 2 diabetic patients, which is more than twice as prevalent as in non-diabetics (*Farooqi et al., 2022*).

Several factors contribute to the development of depression in diabetes. There are common pathophysiological factors associated with depression, and diabetes adds social and psychological factors such as the financial burden of treatment, fear of disability due to the disease, and the strict demands of dietary control or exercise requirements (*Petrak et al., 2015*). Patients with diabetes who developed depression as a comorbidity often present poorer long-term glycemic control (*Heckbert et al., 2010*), nonadherence to medical therapy (*Lin et al., 2006*), and poorer metabolic control (*Egede & Ellis, 2010*).

The comorbidity between diabetes and depression has been recognized as an emerging global challenge (*Fisher et al., 2012*). Thus, studying the pathogenesis of depression in T2DM is of great importance. The role of bile acids (BAs) in the development of many diseases has become a topic of intense interest. Bile acids are essential signaling molecules that can function in various tissues throughout the body and are involved in the regulation of glucolipid metabolism and homeostasis (*Perino & Schoonjans, 2022*). The bile acid receptor FXR (or NR1H4) and other nuclear receptors activated by ligands (pregnancy X receptor, vitamin D receptor, and the membrane receptor TGR5) are all involved in bile acid signaling (*Sun et al., 2018*).

There is evidence that the development of diabetes or depression is linked to the bile acid signaling pathway. On the one hand, serum total bile acid (TBA) levels rise in patients with type 2 diabetes compared to those without the disease (*Sonne et al., 2016*). Serum bile acid concentrations were found to be significantly higher in patients with type 2 diabetes than non-diabetics and were positively associated with insulin resistance (*Sun et al., 2016*). The postprandial plasma concentrations of individual bile acids and FGF19 were significantly increased in type 2 diabetic patients compared to non-type 2 diabetic patients (*Sonne et al., 2016*). Bile acids regulate glucose homeostasis by directly targeting FXR and TGR5 in the intestine, liver, and pancreas, promoting FXR, and inducing intestinal FGF15/19. FXR inhibits hepatic glycolysis and lipogenesis and reduces postprandial glucose utilization (*Caron et al., 2013*; *Duran-Sandoval et al., 2005*;

*Watanabe et al., 2004b*). FXR and FGF15 play an essential role in the development of diabetes (*Yan et al., 2021*).

On the other hand, in recent years, it has become increasingly apparent that bile acids influence brain function during normal physiologic and pathologic conditions (*Mertens et al., 2017*). Even though bile acids can be locally synthesized in the brain (*Kiriyama & Nochi, 2019*), brain bile acids are mainly derived from systemic circulation (*Monteiro-Cardoso, Corliano & Singaraja, 2021*). They are mainly metabolites of the liver and gut microbiota. Therefore, bile acids in the systemic circulation can directly or indirectly influence central processes and thus participate in the neuropathological processes of depression (*Lirong et al., 2022*).

Chronic stress-induced depressed rats had an elevated serum glycocholic acid (GCA) level, but a decreased cholic acid (CA) level was compared with normal control rats (*Zhu et al., 2020*). In a clinical study of patients with Crohn's disease (CD), a positive correlation between GCA and both Zung's Self-Assessment Scale for Anxiety (SAS) and the Self-Rating Scale for Depression (SDS) was observed (*Feng et al., 2022*). However, inconsistent results have also been reported, with one study demonstrating a significant increase in blood cholic acids (CAs) in a mouse model of depression induced by chronic restraint (*Zhang et al., 2010*). The study revealed that FXR altered blood bile acid concentration and composition in the brain *via* FXR knockout mice, contributing to the homeostatic regulation of multiple neurotransmitter systems in different brain regions and modulating neurobehavior (*Huang et al., 2015*). Expression of FXR was significantly downregulated in the medial prefrontal cortex of mice that were subjected to chronic social defeat stress (*Bao et al., 2021*). This work suggested a role for FXR in developing depression and may be a potential new therapeutic target.

Notwithstanding, the profile of BAs and their receptor FXR/FGF15 signaling pathway in depression in T2DM remains largely unknown. This study aims to investigate potential BA-associated molecular mechanisms underlying the disorder of glucolipid metabolism aggravated with chronic stress in diabetic mice.

## MATERIALS & METHODS

### Animals
Male 5-week-old C57BL/6 mice (Animal Certificate No.: SCXK (Su) 2020-0009 from Jiangsu Huachuang Sino Pharma Tech Could., Ltd., Jiangsu, China) were used in the study. All animals were raised in a standardized animal room ($22 \pm 2$ °C, turning on lights from 6 a.m.~6 p.m.), with free access to clean boiled water and rodent chow.

### The high fed streptozotocin mice model
After two weeks of acclimation, the C57BL/6 mice were randomly assigned to two groups using a random number table; one group was fed a chow diet (D12450J, 10% fat, 70% carbohydrate, 20% protein, purchased from Jiangsu Xietong Pharmaceutical Bio-engineering Co., Ltd., Jiangsu, China). The other group was fed a high-fat diet (D12492, 60% fat, 20% carbohydrate, and 20% protein, purchased from Jiangsu Xietong Pharmaceutical Bio-engineering Co., Ltd., Jiangsu, China). After four weeks, each mouse

in the high-fat diet feeding group was given two intraperitoneal doses of streptozotocin (STZ, Sigma-Aldrich Co., St. Louis, Missouri, USA) (80 mg/kg/d with an interval of two days) within 15 min of dissolution according to the adapted protocol (*Furman, 2021*). The STZ powders were prepared immediately with a 0.1 mol/L, pH 4.5 sodium citrate-citric acid buffer (Shanghai Yuanye Bio-Technology Co., Ltd., Shanghai, China). The diabetic status of the mice was ascertained 10 days after the last STZ administration by quantifying blood glucose levels using a Contour plus blood glucose monitoring system (Bayer, Leverkusen, Germany) *via* a blood sample obtained from the tail vein. Mice with a blood glucose concentration higher than 13.9 mmol/L were considered diabetic and were used to establish a model of chronic unpredictable mild stress (CUMS).

## Chronic unpredictable mild stress protocol and experimental designs

A model of depression was developed in which diabetic mice meeting blood glucose requirements were included. This model was used by CUMS to induce a depressive-like phenotype in the C57BL/6 mouse. The CUMS protocol was adapted from the protocol (*Nollet, 2021*). Briefly, the protocol lasted for four weeks after the establishment of the T2DM model, with stressors randomly assigned. The stressors included: damp sawdust for 24 h; no sawdust for 24 h; cage tilting for 24 h; restraint stress for 2 h; cycle disturbances; swimming at 4 °C for 5 min; cage vibration for 20 min; and noise interference for 1.5 h. During the stress procedure, two unexpected stressors were delivered per day, and no single stressor was delivered consecutively for two days. The schedule of the stressors is shown in Table S1. Every mouse subjected to CUMS was housed separately in a single cage, while others were housed in groups. After four weeks of CUMS, the mice were divided into four groups: Control, CUMS, T2DM, and T2DM+CUMS ($n = 10$ per group). Fifty mice were used for this study, and forty were included. Ten mice were excluded, including eight that did not meet the blood glucose requirements and another three that were excluded because of technical failure during CUMS treatment.

## Validation of CUMS mice model
### Open field test
Mice ($n = 10$ per group) were placed in the open-field chambers with a video capture system ($50 \times 50 \times 40$ cm$^3$, L × W × H) from the same position. The Tracking Master software V3.0 (Beijing Zhongshi Dichuang Technology Development Co., Ltd., Beijing, China) was launched to automatically record the activity of the mice within the chamber for 5 min. Chambers were sprayed with 75% ethanol following each mouse trial to avoid odor interference. The room light and noise levels were kept consistent throughout the experiment.

### Tail suspension test
Mice ($n = 10$ per group) were secured to the hook using medical tape at 3/4 of the tail and suspended in reverse suspension. The hanging mouse was approximately 30 cm from the chamber's bottom, with a camera positioned at the level of the hanging apparatus. The Tracking Master V3.0 software (Beijing Zhongshi Dichuang Technology Development

Co., Ltd., Beijing, China) was launched, and the immobility time of the mice was automatically recorded for the final four minutes of the six minute test.

### Forced swimming test

Mice ($n = 10$ per group) were placed in a cylindrical bucket with a diameter of 10 cm and a height of 25 cm. The bucket's water level was such that the mice could stretch their whole body without their tail touching the bottom of the bucket (the water temperature was 23~25 °C, and the water level was 15 cm high). The Tracking Master V3.0 software (Beijing Zhongshi Dichuang Technology Development Co., Ltd., Beijing, China) was launched and the immobility time of each mouse was recorded during the last four minutes of the five minute test (the mice were considered immobile when they were floating on the water surface and their limbs were not moving or their limbs were slightly paddling).

### Sucrose preference test

Each animal ($n = 10$ per group) received two identical bottles during the experiment: one containing clean boiled water and the other containing a solution of 1% sucrose. At the end of the 15 h trial, fluid consumption was measured. Sucrose preference proportion = sucrose solution consumption /(sucrose solution consumption + boiled water consumption) × 100%.

## Sacrifice and samples collection

After completion of behavioral tests, mice were sacrificed under anesthesia using 4% chloral hydrate. The blood was collected, coagulated at room temperature for 30 min, and centrifuged at 3,000 rpm for 15 min.

The supernatant serum was separated and divided into multiple vials before being stored at −80 °C. Following cardiac perfusion, the liver and ilea were dissected and snap-frozen in liquid nitrogen. The liver was weighed before being cut into two parts, one fixed with cold 4% paraformaldehyde and the other flash frozen and stored at −80 °C, while the pancreas was fixed with cold 4% paraformaldehyde. Serum samples were analyzed for serum lipids, hepatic function, and insulin levels. Western blots were performed on the frozen liver and ileum. The fixed livers and pancreas were used for immunohistochemistry.

## Serum lipids and liver function

Serum levels of glutamic oxaloacetic transaminase (ALT), glutamic alanine transaminase (AST), total cholesterol (CHO), triglycerides (TG), high-density lipoprotein cholesterol (HDL-C), low-density lipoprotein cholesterol (LDL-C), non-esterified fatty acid (NEFA), and blood glucose (GLU) were measured by a Beckman Coulter biochemical analysis system (AU5800).

## Enzyme-linked immunosorbent assays

The concentration of glycated hemoglobin A1c (GHbA1c) in the lysate was measured using the Mouse Glycated Hemoglobin A1c (GHbA1c) ELISA Kit (#CSB-E08141m; CUSABIO, Wuhan, China, https://www.cusabio.com/) and the Mouse Insulin ELISA

kit (#KE10089; Proteintech Group, Inc., Wuhan, China) following the manufacturer's instructions in order to determine the red blood cell and serum insulin levels.

## Western blotting

Total protein was extracted from liver and ileum specimens using RIPA ($n = 3$ per group), and the extracts were used to detect FXR, SHP, and FGF15 protein expression. The protein concentration was determined using the BCA method (Yoche Biotechnology, Shanghai, China), and proteins were denatured sequentially using gel electrophoresis and wet-membrane blotting. After blocking for 2 h at room temperature, diluted primary antibodies of FXR (1:1000, Cell Signaling Technology, Inc., Boston, USA), FGF15 (1:1000, Santa Cruz Biotechnology, Inc., California, USA), SHP (1:500, ABclonal Technology Co., Ltd., Wuhan, China), GAPDH (1:10,000; ABclonal Technology Co., Ltd., Wuhan, China), and β-actin (1:1000; ImmunoWay Biotechnology, Plano, TX, USA) were added for overnight incubation at 4 °C. Following the recovery of the primary antibodies, the membrane was washed three times with TBST (Sangon Biotech, Shanghai, China) for 10 min each. Then, the secondary antibodies (1:5,000; ImmunoWay Biotechnology, Plano, TX, USA) were added to the membrane and were incubated for 1 h at room temperature. After washing the membrane, the ECL reagent was used for image development, and the protein bands were analyzed using the Image J software.

## Total RNA extraction and quantification by qPCR

Total RNA was extracted from the mouse livers and ileum using the TRIzol reagent (Thermo Fisher Scientific, Waltham MA, USA) ($n = 3$ per group). Subsequently, the products were reverse transcribed using the Evo M-MLV RT Mix Kit with gDNA Clean for qPCR (AG11728; Accurate Biotechnology, Hunan, Co., Ltd., Hunan, China). qPCR with reverse transcription was performed using the SYBR Green Premix Pro Taq HS qPCR Kit (AG11718; Accurate Biotechnology, Hunan, Co., Ltd., Hunan, China) on Applied Biosystems Quant Studio 6 Flex apparatus (Thermo Fisher Scientific, Waltham, MA, USA). The Ct values were calculated using the $2^{-\Delta\Delta Ct}$ method in order to analyze the relative level of gene expression level. Table S2 shows the forward and reverse sequences of the qPCR primers that were designed and synthesized (Sangon Biotech, Shanghai, China). All the procedures were carried out following the manufacturer's instructions. Target gene values were normalized to ACTB, and the relative expression levels were displayed as fold changes relative to the control group values.

## Histopathological analysis

For morphological examination, dehydrated liver specimens ($n = 3$–4 per group) were embedded in paraffin, cut into 4~6 µm sections, stained with H&E, and processed for analysis. Oil Red O staining revealed the deposition of lipids in the liver. Frozen liver sections were air-dried at room temperature for 2 h before fixation for 5 min with 4% paraformaldehyde. Sections were washed and rinsed in 60% isopropanol to remove excess water. Afterward, the sections were incubated in Oil Red O reagents for 10 min before removal from the solution with 60% isopropanol. Nuclei were stained with hematoxylin for 5 min before being washed with water.

## Statistical analysis

Statistical analysis was performed using GraphPad 9.3.0 software (GraphPad Software, La Jolla, CA, USA). Experimental data were presented as the mean ± standard error (SEM). All data were analyzed with unpaired two-tailed Student's $t$-test (for data from two groups) and one-way ANOVA followed by Tukey's multiple comparison post hoc test (for data from more than two groups). A difference of $P < 0.05$ indicates a statistically significant difference.

## RESULTS

### Type 2 diabetic mice with depressive-like phenotype comorbidity model validation

After acclimation and randomization, half of the mice (20 out of 40) in the experimental group were subjected to a 4-week high-fat chow intervention and STZ *via* intraperitoneal injection to create a model of T2DM. Subsequently, mice from the diabetic model that met glycemic requirements were included in the 4-week CUMS process to establish a depressive-like phenotype. Both the T2DM and T2DM+CUMS groups continued to be fed high-fat diets during CUMS (Fig. 1A).

Behavioral analysis was used to verify whether the depressive-like phenotype was successfully induced in diabetic mice. The open-field test revealed that the distance and duration of activity in the central zone were significantly shorter in the CUMS group compared to the Control group. Moreover, the T2DM+CUMS group showed a shorter distance and duration of activity in the central zone and a minor total distance covered compared to the T2DM group (Fig. 1B). CUMS mice exhibited longer immobility time in the tail suspension and forced swim tests compared to the Control group. Mice in the T2DM+CUMS group also showed significantly longer immobility time for tail suspension and forced swim compared to the T2DM group (Figs. 1C and 1D). CUMS or T2DM mice exhibited a lower percentage of sucrose preference relative to the Control group, while T2DM+CUMS showed a lower level of sucrose preference compared to CUMS (Fig. 1E).

### Type 2 diabetic mice with depressive-like phenotype in glucolipid metabolism

At the end of the CUMS protocol, *in vivo* metabolic measurements, including body weight, fasting blood glucose, and serum lipids (CHO, TG, LDL-C, HDL-C, NEFA), were taken to evaluate the effect of depression in T2DM on glucolipid metabolism. All of the stressed mice had significantly lower body weight gains and unstable blood glucose levels (Figs. 2A and 2C). Body weight was not significantly different between the four groups of mice, but the body weights of the mice in the T2DM group were slightly greater than those in the T2DM+CUMS group (Fig. 2B). Fasting blood glucose was measured after the CUMS model was established, and it was found that T2DM mice had higher levels of blood glucose than the Control mice. In contrast, there were no significant differences between mice in the T2DM+CUMS group and those in the T2DM group (Fig. 2D). A comparison of serum lipids revealed lipid disorders among these groups. T2DM was related to higher serum CHO, LDL-C, HDL-C, and NEFA levels in the serum (Figs. 2E

**A**

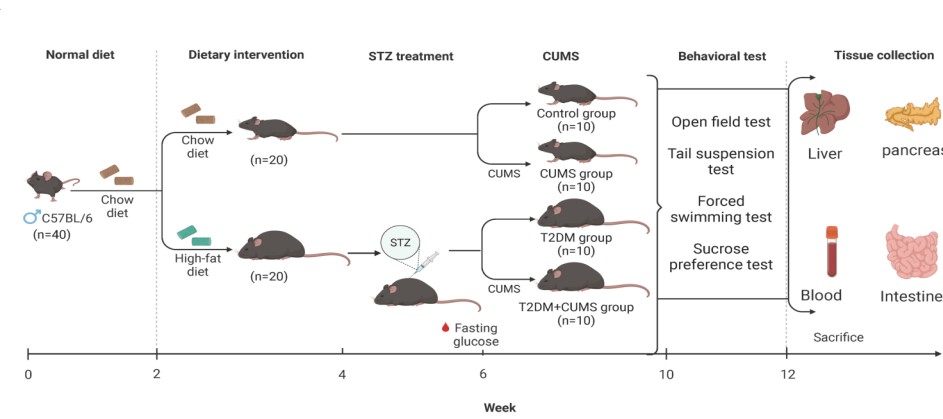

**B**

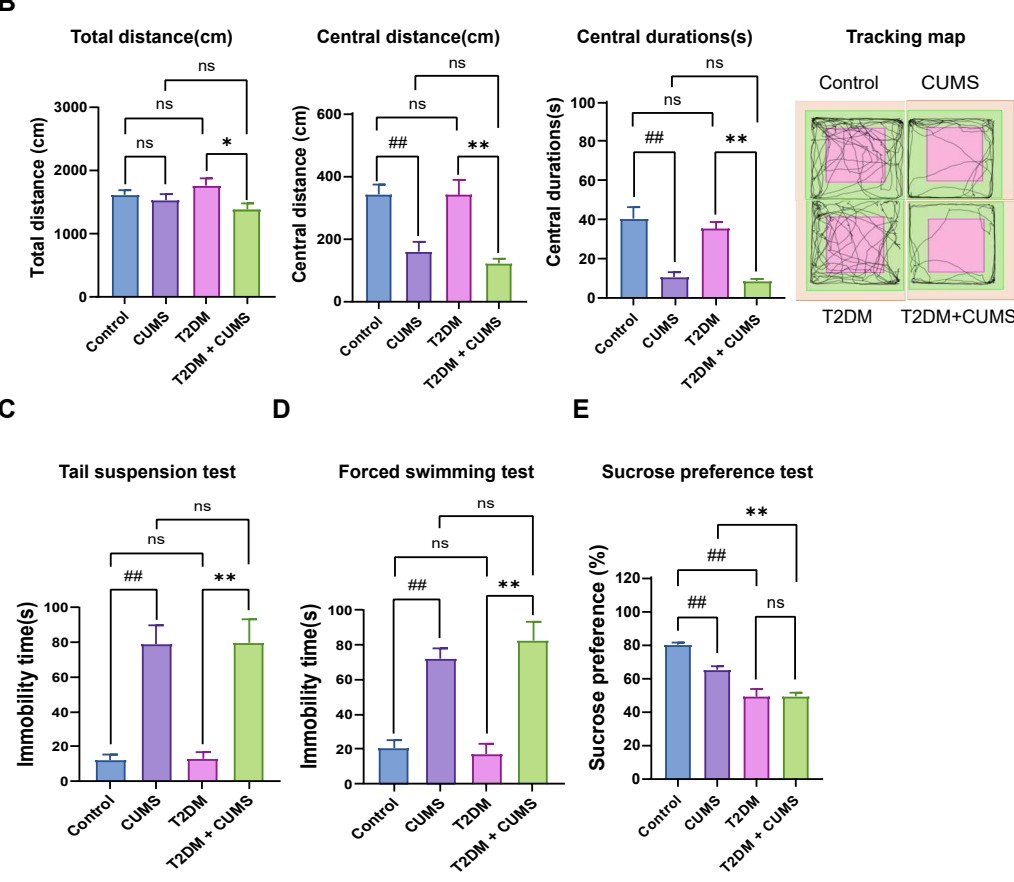

**Figure 1** **Type 2 diabetic mice with the depression-like phenotype comorbidity model validation.** (A) Schedule of T2DM and chronic stress-induced depression-like phenotype comorbidity model establishment. The figure was created with BioRender.com. (B) Total distance, central distance, and central durations in open field test and the tracking map. (C) Immobility durations in tail suspension test. (D) Immobility durations in the forced swimming test. (E) Sucrose preference rate in the sucrose preference test. Data presented as mean ± SEM, $n = 10$ per group. $^{\#\#}P < 0.01$ compared with the Control group, $^*P < 0.05$, $^{**}P < 0.01$ compared with the T2DM+CUMS group. Image credit: Beijing Zhongshi Dichuang Technology Development Co., Ltd. Created with BioRender.com.

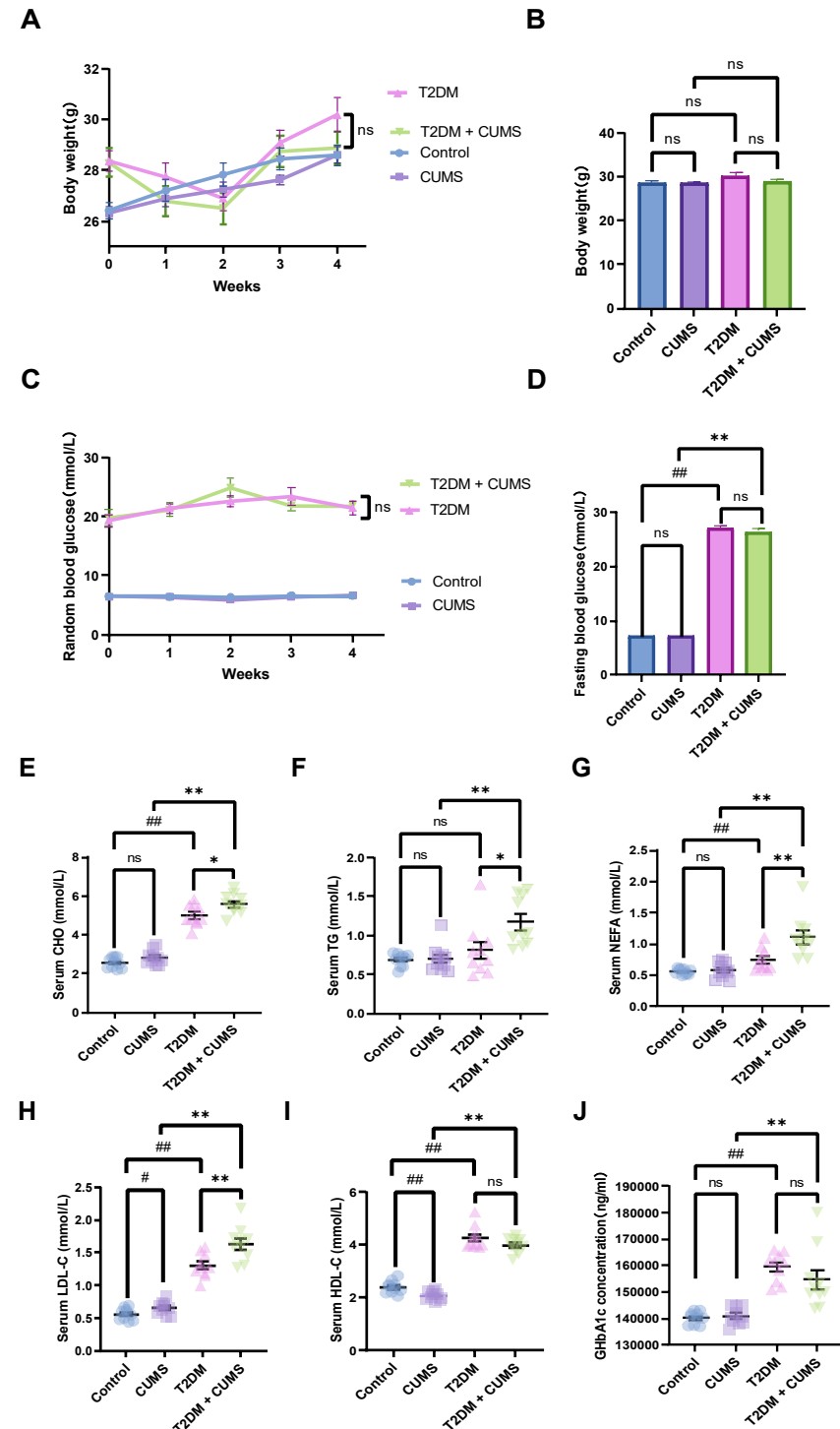

**Figure 2  Type 2 diabetic mice with the depression-like phenotype in glycolipid metabolism.** (A) Body weight weekly in CUMS procedures. (B) Body weight in mice after CUMS procedures. (C) Random blood glucose weekly in CUMS procedures. (D) Fasting blood glucose after CUMS procedures. (E) Serum CHO in mice. (F) Serum TG in mice. (G) Serum NEFA in mice. (H) Serum LDL-C in mice. (I) Serum HDL-C in mice. (J) GHbA1c in mice. Data presented as mean ± SEM, $n = 10$ per group, $^{\#}P < 0.05$, $^{\#\#}P < 0.01$ compared with the Control group, $^{*}P < 0.05$, $^{**}P < 0.01$ compared with the T2DM+CUMS group.

and 2G–2I). Serum CHO, TG, LDL-C, and NEFA levels were significantly higher in the T2DM+CUMS mice compared to the T2DM mice (Figs. 2E–2H). Concerning GHbA1c testing, T2DM+CUMS mice had higher levels of GHbA1c compared to CUMS mice but did not differ significantly when compared with the T2DM mice (Fig. 2J).

### Insulin-related indicators and histopathological analysis of the pancreas in type 2 diabetic mice with a depressive-like phenotype

As stated, the serum glucose levels in the CUMS and T2DM mice were greater than those in the Control group. In addition, serum glucose levels were higher in the T2DM+CUMS group compared to the CUMS or T2DM group (Fig. 3A). More importantly, in the insulin assay, it was readily discovered that under fasting conditions, CUMS mice exhibited a significant increase in fasting insulin (Fig. 3B). Meanwhile, the T2DM+CUMS group showed a slight increase compared to the T2DM group (Fig. 3B). Despite the lack of statistical significance, the HOMA-IR index was increased in stressed mice, particularly in the CUMS and T2DM groups when compared with Control mice. HOMA-IR was also higher in T2DM+CUMS mice compared with T2DM mice (Fig. 3C). In terms of HOMA-β, the HOMA-β index was lower in the T2DM group than in the Control group, but there was no significant difference in the HOMA-β index of the T2DM+CUMS group when compared with the T2DM group (Fig. 3D).

Pancreatic histomorphology revealed that the islets of the Control group mice were round or oval cell clusters of various sizes, with complete and regular structures, and clear edges. The cell clusters were scattered among the pancreatic vesicles with more cell clusters and abundant β-cells in the islets; they were uniform in size, full and tightly arranged. However, the islets of the CUMS group mice were swollen and deformed, with light cytoplasmic staining or vacuolation. In the T2DM group, there was a decrease in the number of islet cell clusters, as well as a shrunken islet area, irregular morphology, unclear islet border, disorganized structure, swollen and detorted islet cells, clear or vacuolated cytoplasmic staining, and obvious nuclear consolidation or nuclear loss. Some of the exocrine gland vesicles were found deep within the islet, while in the T2DM+CUMS group, the area of islets was severely shrunken, the morphology was irregular, the islet cells were swollen and detorted, and the cytoplasmic staining was mild or vacuolated when compared with T2DM mice (Fig. 3E).

### Liver function, total bile acids, and histopathological analysis of the liver in type 2 diabetic mice with depressive-like phenotype

Through liver pathology analysis, it was observed that the comorbidity of type 2 diabetes mellitus and depressive-like phenotype exacerbated liver steatosis and lobular inflammation (Figs. 4A–4C). In the Control group, hepatic lobules and cords were irregularly arranged, hepatocytes displayed unequal sizes, and some were cloudy and swollen, with a small amount of inflammatory cell infiltration in the confluent area. In the CUMS group, hepatic lobules and cords were also irregularly arranged, hepatocytes were unequal in size, some were cloudy and swollen with partial loss of nuclei, and inflammatory cells infiltrated the confluent area, indicating moderate lesions. In T2DM mice, hepatic lobules and cords were irregularly arranged, hepatocytes displayed unequal sizes, with

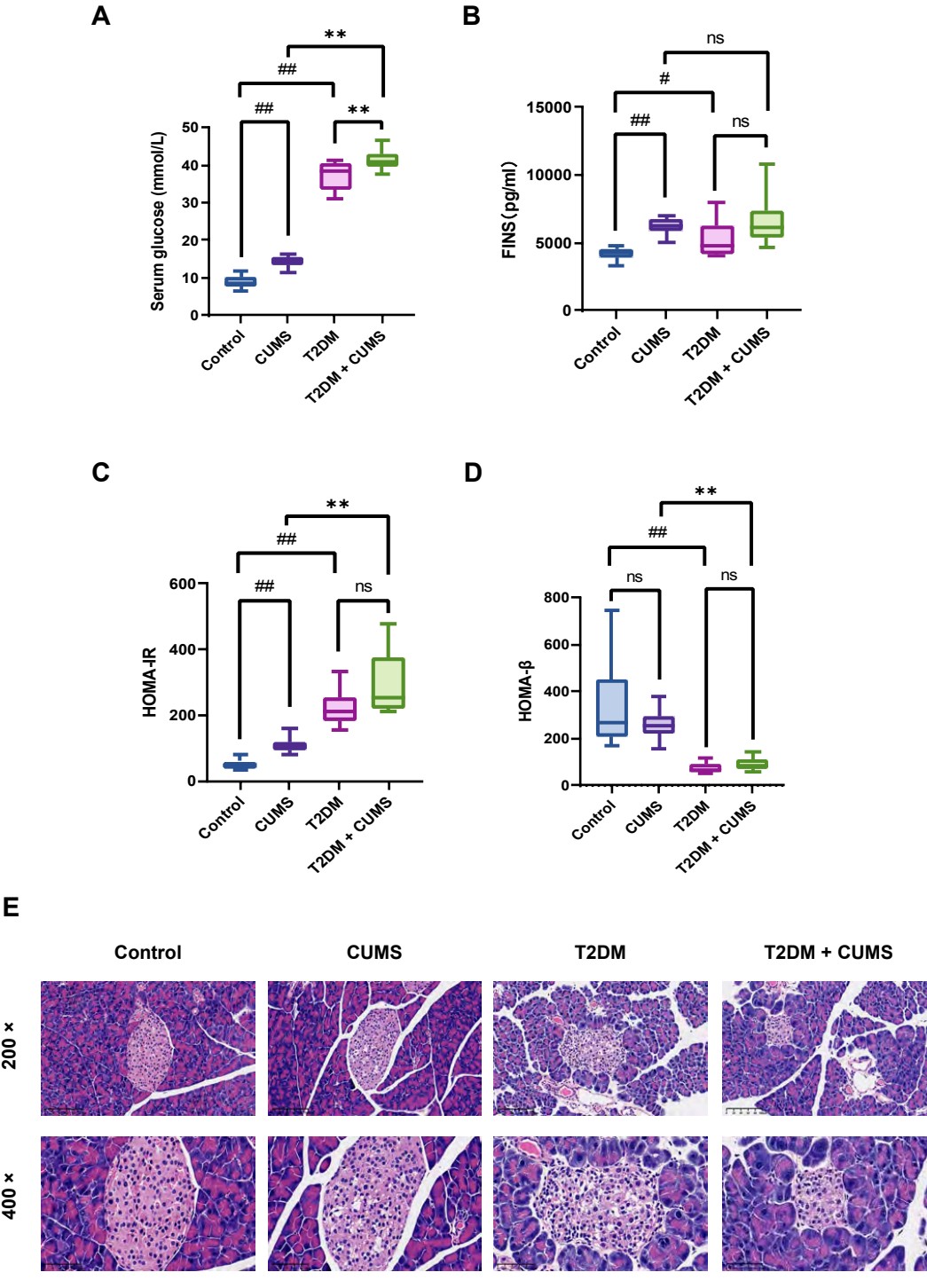

**Figure 3** **Insulin-related indicators and histopathological analysis of the pancreas in type 2 diabetic mice with depression-like phenotype.** (A) Serum glucose in mice. (B) FINS in mice. (C) HOMA-IR index in mice. (D) HOMA-β index in mice. (E) Representative images of pancreas sections from four groups of mice stained with H&E. Data presented as mean ± SEM, $n = 10$ per group. $^{#}P < 0.05$, $^{##}P < 0.01$ compared with the Control group, $^{**}P < 0.01$ compared with the T2DM+CUMS group. Images were taken at $200\times$ or $400\times$ magnification.

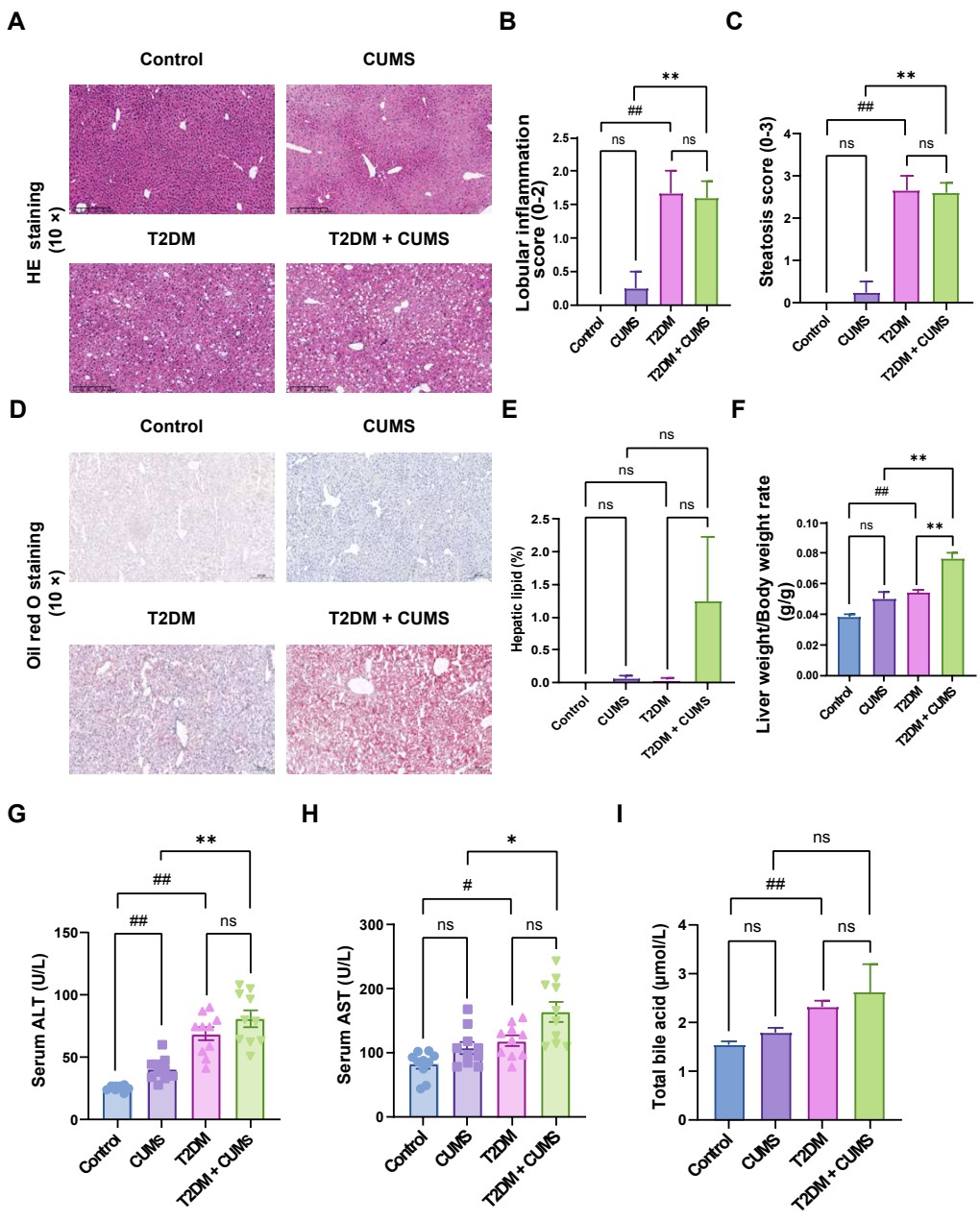

**Figure 4  Liver function, total bile acids and histopathological analysis of the liver in type 2 diabetic mice with depression-like phenotype.** (A) Representative images of liver sections from four groups of mice stained with H&E. (B) Lobular inflammation score in four groups. (C) Steatosis score in four groups. (D) Representative images of liver sections from four groups of mice stained with Oil Red O. (E) Hepatic lipids percentage in four group. (F) Liver weight/body weight ratio in mice. (G) ALT level in mice. (H) AST level in mice. (I) Total bile acid level in mice. Data presented as mean ± SEM, $n = 10$. [#] $P < 0.05$, [##] $P < 0.01$ compared with the Control group, [*] $P < 0.05$, [**] $P < 0.01$ compared with the T2DM+CUMS group. Images were taken at 10× magnification.

some degeneration, and balloon-like changes were visible, indicating mild to moderate steatosis. Additionally, the T2DM+CUMS group exhibited unequally arranged lobules and hepatic cords, as well as hepatocytes that were degenerated and ballooned, indicating moderate to severe steatosis (Fig. 4A). Further investigation of the effect of comorbidity on liver steatosis and inflammation revealed no significant difference in inflammation and steatosis scores between the T2DM+CUMS group and the T2DM group. However, comorbidity led to increased lipid accumulation in the liver, as evidenced assessed by Oil red O staining (Figs. 4D and 4E). The comorbidity increased the ratio of liver weight to body weight in mice from both the CUMS group and the T2DM group (Fig. 4F). ALT and AST levels in the liver were also elevated in response to the comorbidity (Figs. 4G and 4H). Furthermore, total bile acid levels were higher in the T2DM+CUMS group compared to the T2DM group (Fig. 4I). Taken together, these findings suggested that co-morbidity exacerbated problems with liver function, hepatic steatosis and inflammation, and resulted in an increase in total bile acids in T2DM.

### T2DM and depressive-like phenotype regulated by FXR/SHP/FGF15

The hepatic FXR mRNA level was slightly lower in the T2DM+CUMS group, whereas the SHP mRNA level was slightly higher in the liver when comparing the T2DM group to the T2DM+CUMS group (Figs. 5A and 5B). Similarly, the T2DM+CUMS group exhibited a lower protein level of hepatic FXR but showed an increasing tendency in the protein level of SHP in the liver (Fig. 5C). In terms of the ileum, the T2DM+CUMS group displayed a slightly lower expression of ileal FXR mRNA compared to the T2DM group. Additionally, the T2DM+CUMS group had significantly lower ileal FGF15 mRNA expression compared to the T2DM group (Figs. 5D and 5E). Similarly, the ileal FXR protein level was significantly lower in the T2DM+CUMS group compared to the T2DM group, and a lower ileal FGF15 protein level was also observed in the T2DM+CUMS group (Fig. 5F). These results suggested that depression in T2DM may impact the regulation of FXR, SHP, and FGF15 in the liver and ileum, potentially contributing to the observed metabolic changes and liver dysfunction. Further investigation is warranted to fully elucidate the underlying mechanisms.

## DISCUSSION

Comorbidities between mental and physical diseases pose a significant challenge in healthcare. Depression and diabetes are prime examples of mental/physical comorbidities, and their prevalence is on the rise, mainly due to increasing life expectancy and various other factors. Patients with diabetes are more susceptible to developing depression during their illness, and specific characteristics are associated with this comorbidity. These include being female, younger age or older age, living alone, having poor social support, having low socioeconomic status, experiencing sleep problems, and lacking physical exercise and a balanced diet (*Agardh et al., 2011*; *Bădescu et al., 2016*; *Sartorius, 2018*). Moreover, growing evidence also suggests that depression and type 2 diabetes share a common biological origin, mainly related to innate immune overactivation and

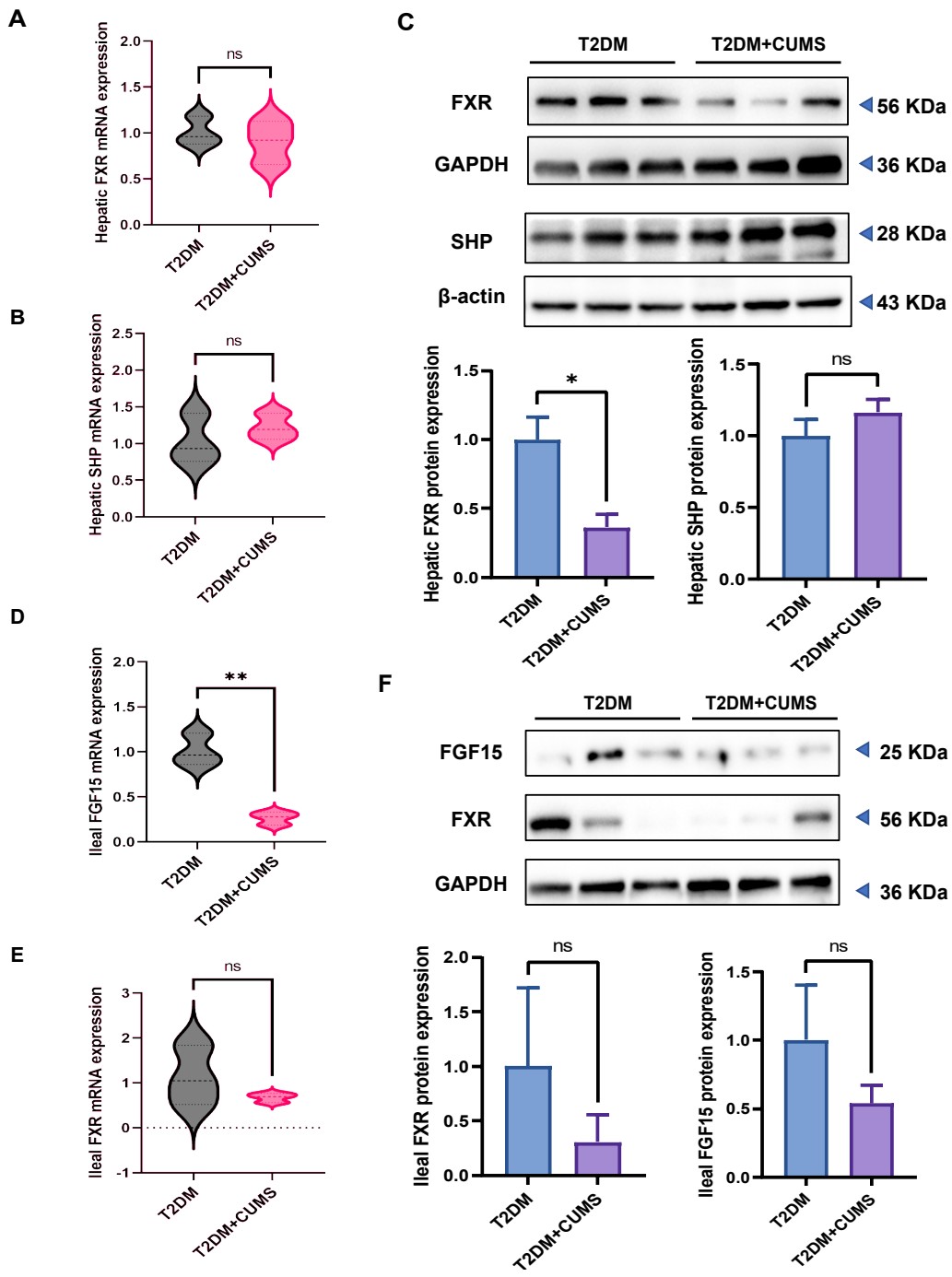

**Figure 5** **T2DM and depression-like phenotype regulated by FXR/SHP/FGF15.** (A) Relative expression of FXR mRNAs in the livers of mice. (B) Relative expression of SHP mRNAs in the livers of mice. (C) Relative expression of FXR and its target gene SHP proteins in the livers of mice.(D) Relative expression of FGF15 mRNAs in the ileum of mice. (E) Relative expression of FXR mRNAs in the ileum of mice.(F) Relative expression of FXR and its target gene FGF15 proteins in the ileum of mice. Data presented as mean $\pm$ SEM. * $P < 0.05$, ** $P < 0.01$ compared with the T2DM group.

cytokine-mediated inflammatory responses that may be regulated by the hypothalamic-pituitary-adrenal axis (*Moulton, Pickup & Ismail, 2015*). Additionally, the development of depression in diabetes has been linked to dysregulation of bile acid synthesis and metabolism in both rodents and humans (*Feng et al., 2022*; *Lirong et al., 2022*; *Sonne et al., 2016*; *Sun et al., 2016*; *Zhu et al., 2020*). The bile acid nuclear receptor FXR is an essential component of the liver-gut-brain axis that has also been shown to play a vital role in diabetes and depression (*Bao et al., 2021*; *Caron et al., 2013*; *Duran-Sandoval et al., 2005*; *Huang et al., 2015*; *Watanabe et al., 2004b*; *Yan et al., 2021*). The association between diabetes and depression is robust, and the development of comorbidities may link to the bile acid nuclear receptor FXR. Understanding these connections is vital for better managing and treating patients with these comorbid conditions. Further research is needed to explore the underlying mechanisms fully and develop effective interventions.

While studies have shown that the bile acid nuclear receptor FXR may play an essential role in diabetes or depression, there are fewer in-depth studies on the liver-gut axis FXR and its target genes SHP and FGF15 regulating depression in T2DM. Therefore, we set out to elucidate the role of FXR and its target genes SHP and FGF15 developing of diabetic mice aggravated by chronic stress.

The C57BL/6 mouse strain is susceptible to the metabolic effects of a high-fat diet, making it a straightforward model to employ (*Luo et al., 1998*). In addition, the use of STZ to stimulate a reduction β cell number, further mimicking T2DM, has been previously explored (*Gilbert, Fu & Liu, 2011*; *Podrini et al., 2013*; *Shao et al., 2014*). In this study, we successfully established a T2DM mouse model using a 4-week high-fed diet in combination with STZ (i.p.). After confirming fasting blood glucose levels, T2DM mice were subjected to the establishment of CUMS. Behavioral tests validated the efficacy of the CUMS model, and the T2DM mouse model with a depressive-like phenotype was successfully established. The sucrose preference test revealed that T2DM mice had a lower sucrose preference rate than the Control group, indicating that T2DM mice developed a depressive-like phenotype during the disease, as previously reported (*Hai-Na et al., 2020*). Furthermore, we found that CUMS exacerbated the depressive-like phenotype in diabetic mice. The diabetic mice developed a depressive-like phenotype, which further worsened with continued exposure to CUMS.

We identified a bidirectional predisposing risk between T2DM and depression in a 4-week CUMS mouse model. Serum samples from the mice were analyzed to look for relevant biochemical indicators. Indeed, chronic stress elevated the levels of CHO, TG, HDL-C, LDL-C, and NEFA levels in diabetic mice, while diabetes also increased the levels of CHO, TG, LDL-C, HDL-C, and NEFA in stressed mice. It is widely recognized that diabetic mice exhibit a disturbed serum lipid profile, characterized by higher levels of TC, TG, and LDL-C, and lower levels of HDL-C (*Manickam et al., 2022*). However, the lipid profiles of diabetic mice are not consistently reported in the literature. Interestingly, our study observed that diabetic mice exhibit elevated HDL-C levels, a finding that aligns with some studies (*Ivanovic et al., 2015*; *Qin et al., 2018*).

Insulin resistance and β cell failure are critical characteristics of T2DM (*Defronzo, 2009*). Insulin resistance is characterized by a diminished response of peripheral tissues

to insulin stimulation, leading to elevated peripheral insulin levels (*Polidori et al., 2022*). It has a bidirectional causal interaction with depression. Insulin resistance increases the risk and severity of depression; conversely, depression increases the risk and severity of insulin resistance (*Watson et al., 2018*). A meta-analysis study found that insulin levels and the HOMA-IR index were higher in patients with acute depression (*Fernandes et al., 2022*). In our study, stressed mice in the CUMS or T2DM+CUMS groups developed insulin resistance. The primary physiological function of β-cells is the synthesis and secretion of insulin. When pancreatic β-cells become dysfunctional and insulin secretion is insufficient, blood glucose levels rise dramatically, leading to the gradual development of diabetes. Normal pancreatic β-cells are highly sensitive to changes in blood glucose levels, and a decrease in the number of pancreatic β-cells, whether direct or indirect, can directly lead to impaired insulin secretion and β-cell function (*Rorsman & Ashcroft, 2018*). The comorbidity HOMA-β index was found to be lower in the T2DM+CUMS in our studies. Moreover, the number and area of pancreatic islet β-cells were reduced. The gold standard for assessing the concentration of glycemic control is GHbA1c, reflecting the average blood glucose level over the previous 8–12 weeks. Our results showed that comorbid CUMS did not increase the concentration of GHbA1c.

Interestingly, we observed decreased levels of HDL-C and GHbA1c in T2DM mice with depression compared to diabetic mice, although these differences were not statistically significant. We hypothesized that the lower levels of HDL-C and GHbA1c might be influenced by certain metabolic enzymes associated with lipid accumulation and that exposure to CUMS might play a role in these mechanisms. However, further research confirms these findings.

Based on the results of liver pathology, liver function, liver weight/body weight ratio, and total bile acid level, the T2DM+CUMS group exhibited a more severe degree of conditions such as steatosis and inflammatory cell infiltration compared to the T2DM or CUMS groups. Additionally, the T2DM+CUMS group showed an increased liver/body weight ratio, elevated levels of ALT or AST, and a higher total bile acid level. These findings suggest a disturbance in lipid metabolism following the comorbidity of depression and T2DM.

Bile acids, being potent detergents, can potentially be toxic to cells, necessitating strict control over their cellular levels. It has been found that FXR plays a crucial role in maintaining bile acid homeostasis and may protect cells from bile acid toxicity. FXR, a ligand-activated member of the nuclear receptor superfamily, is primarily expressed in the liver and ileum and plays a vital role in developing metabolic diseases (*Matsubara, Li & Gonzalez, 2013*). FXR activation in the liver has been shown to protect against the development of hepatic steatosis. Hepatic FXR deficiency exacerbates hepatic steatosis in a high-cholesterol diet model (*Schmitt et al., 2015*). FXR activation in the liver reduces hepatic lipid content by decreasing lipogenesis and increasing fatty acid oxidation (*Pineda Torra et al., 2003*; *Watanabe et al., 2004a*). FXR-/- mice exhibited increased liver fat accompanied by elevated triglycerides, cholesterol, non-esterified fatty acid, and lipoproteins (VLDL-C and LDL-C) (*Sinal et al., 2000*). By contrast, in diabetic (db/db), obese (ob/ob), or wild-type mice, the use of bile acids or FXR agonists reduces

plasma triglycerides, fatty acids, and cholesterol, as well as reducing hepatic lipid/steatosis (*Watanabe et al., 2004a*; *Zhang et al., 2006*). In the present study, we observed reduced hepatic FXR mRNA and protein expression in T2DM+CUMS animal models compared to T2DM mice, suggesting a potential role for FXR in T2DM or CUMS. Reducing sterol regulatory element binding protein 1c(SREBP1c) expression by inducing SHP and FXR activation leads to a decreased expression of adipogenesis-related genes (*Watanabe et al., 2004a*). We found that hepatic FXR protein and mRNA levels were downregulated in T2DM+CUMS groups of mice, while the protein of SHP, the target gene of FXR, in the liver, was not significantly different among the T2DM and T2DM+CUMS groups. As opposed to FXR, SHP showed increased mRNA and protein levels in T2DM+CUMS group compared to the T2DM group. Previous studies have shown that liver-specific SHP deletion prevents hepatic steatosis and fatty liver development (*Akinrotimi et al., 2017*). Evidently, CUMS exacerbates hepatic steatosis in diabetic mice, possibly through the FXR-SHP pathway. This suggests that the disturbance in lipid metabolism in diabetic mice following the administration of chronic stress-induced depressive-like phenotype may be mediated by the FXR-SHP pathway.

In addition to regulating hepatic lipid levels, FXR also influences hepatic glucose metabolism. This suggests a complex interplay between glucocorticoids and bile acid homeostasis. From this perspective, a complex interplay between glucocorticoids and bile acid homeostasis is conceivable. FXR activation has been shown to reduce gluconeogenesis while increasing glycolysis (*Ma et al., 2006*). An early stage of insulin resistance was observed in FXR-/- mice indicating a role for FXR in glucose metabolism (*Cariou et al., 2006*; *Zhang et al., 2006*). Consistent with this, in wild-type or diabetic db/db or insulin-resistant ob/ob mice, the use of bile acids or FXR-specific agonists increased insulin sensitivity and reduced blood glucose concentrations (*Cariou et al., 2006*; *Zhang et al., 2006*). Our results mirrored these findings, with the downregulation of hepatic FXR protein and mRNA levels observed in both T2DM and T2DM+CUMS groups of mice. Enterocytes can activate the nuclear receptor FXR, leading to the production of FGF15 (*Kliewer & Mangelsdorf, 2015*). Beyond the enterohepatic cycle, FGF15 can signal in an endocrine manner and is involved in lipid and glucose metabolism (*Owen, Mangelsdorf & Kliewer, 2015*). Previous studies have shown that FGF15 secreted from the ileum after FXR activation has insulin-like effects and inhibits hepatic gluconeogenesis (*Kir et al., 2011*; *Potthoff et al., 2011*; *Potthoff, Kliewer & Mangelsdorf, 2012*; *Schaap, 2012*). In the present study, the mRNA and protein expression levels of FGF15 were downregulated in the ileum of both T2DM and T2DM+CUMS groups of mice. Comprehensive studies by *Jia et al. (2019)* determined that the glucocorticoid receptor, the classical receptor of dexamethasone, is highly expressed in the ileum. The activation of glucocorticoid receptors by dexamethasone may be related to attenuating FXR activation, resulting in the downregulation of Fgf15 expression in the ileum.

Only about 20% of BAs in the brain are locally synthesized (*Monteiro-Cardoso, Corliano & Singaraja, 2021*). BAs act as a ligand for the nuclear receptor FXR. Previous studies have found that chronic social defeat stress (CSDS) significantly decreased in FXR expression in the medial prefrontal cortex of depressive-like phenotypic mice (*Bao*

*et al., 2021*). In the present study, the results showed that mice in T2DM+CUMS group exhibited the downregulated expression of hepatic FXR protein and mRNA, while the expression of hepatic SHP protein and mRNA was upregulated. In the ileum, mice in T2DM + CUMS group had downregulated expressions of the FXR protein and mRNA, and the mRNA and protein expression of FGF15 also showed a decreasing trend in the T2DM+CUMS group.

# CONCLUSIONS

To the best of our knowledge, the association of depression in T2DM and the activation of FXR or its target genes has not been reported. This study identifies FXR and its downstream gene, FGF15, as key components of depressive-like phenotypes development and suggests the FXR-FGF15 axis as a potential new therapeutic target for the comorbidity of T2DM and depression.

# ACKNOWLEDGEMENTS

We would like to express our profound gratitude to our colleagues and mentors for their invaluable guidance, insightful critiques, and patient encouragement throughout the course of this research. Their expertise and knowledge have been instrumental in shaping this study.

## Funding

This work was supported by the Shanghai Municipal Science Commission (grant number 19411971700). The funders had no role in study design, data collection and analysis, decision to publish, or preparation of the manuscript.

## Grant Disclosures

The following grant information was disclosed by the authors:
Shanghai Municipal Science Commission: 19411971700.

## Competing Interests

The authors declare there are no competing interests.

## Author Contributions

- Weijia Cai conceived and designed the experiments, performed the experiments, analyzed the data, prepared figures and/or tables, authored or reviewed drafts of the article, and approved the final draft.
- Canye Li performed the experiments, analyzed the data, prepared figures and/or tables, authored or reviewed drafts of the article, and approved the final draft.
- Zuanjun Su performed the experiments, analyzed the data, prepared figures and/or tables, and approved the final draft.

- Jinming Cao performed the experiments, prepared figures and/or tables, and approved the final draft.
- Zhicong Chen performed the experiments, prepared figures and/or tables, and approved the final draft.
- Yitian Chen performed the experiments, prepared figures and/or tables, and approved the final draft.
- Zhijun Guo performed the experiments, prepared figures and/or tables, and approved the final draft.
- Jian Cai conceived and designed the experiments, authored or reviewed drafts of the article, and approved the final draft.
- Feng Xu conceived and designed the experiments, authored or reviewed drafts of the article, and approved the final draft.

## Data Availability

The raw measurements are available in the Supplementary Files.

## Supplemental Information

Supplemental information for this article can be found online at http://dx.doi.org/10.7717/peerj.16407#supplemental-information.

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
