# Peer review of "Profile of the bile acid FXR-FGF15 pathway in the glucolipid metabolism disorder of diabetic mice suffering from chronic stress"

_PeerJ, doi:10.7717/peerj.16407_

## Round 0.1 · original submission · Major Revisions

The manuscript should be revised in the light of the reviewers' comments.
English should be thoroughly re-checked by a professional English speaker.

Reviewer 1 ·

Basic reporting

The format of Table 1 needs to be modified.
The results of PCR and WB in Figure 5 suggest that they can be shown in different colors or forms, they look too repetitive now, and the x-axis of AB and CD in the figure is not uniform.

Experimental design

Did the T2DM and T2DM+CUMS groups continue to be fed with high-fat diets during CUMS?
The body weight of these two groups in Figure 2A decreased rapidly and did not differ from that of the Control and CUMS groups after 4 weeks, was this decrease due to CUMS?
Was there a significant difference in body weight between the T2DM and T2DM+CUMS groups and the other two groups before CUMS?

Validity of the findings

The WB results for FGF15 and FXR in Figure 5F have too much variability between samples. Is it possible to support the conclusion? Please explain or redo it.
The discussion mentioned that BAs and FXR activation in the brain are associated with the nervous system, and since the purpose of this study was to investigate the association of FXR-FGF15 with the development of depression in T2DM combined, why not test the expression of FXR and its downstream targets in the brain?

Reviewer 2 ·

Basic reporting

In this paper, Cai et al. investigated the effects of depression on type 2 diabetes mellitus (T2DM) by accessing to body weight, serum metabolic profiles (levels of glucose, cholesterol, triglyceride, etc.), hepatic inflammation and lipid accumulation, liver damage, by producing and T2DM-mimetic mice and giving chronic unpredictable mild (not mile as written by authors) stress (CUMS). Authors also quantified bile acid production, hepatic FXR and SHP levels as well as ileal FXR and FGF15 levels of those animals to show importance of FXR to gut-liver-brain axis. Their data provide significant alteration of serum metabolic profiles and remarkable elevation in hepatic lipid levels in T2DM+CUMS. Authors further show the reduction of hepatic FXR protein and ileal FGF15 expression, suggesting attenuated of FXR signaling in those tissues when suffering from depression.

The data are of potential interest, particularly in the striking elevation of lipid levels in the T2DM+CUMS liver. However, their manuscript has so many wrong usage of terms that appear to be based on lack of authors’ attention to the manuscript. These are only a small fraction of examples including some major concerns:
1. Ln.25, “This study was aimed….” > “This study aimed”.
2. Ln.28, “…high-fat diet…” > “…high-fat diet feeding…”
3. Ln.31, “CUMS” needs to be spelled out on this line because of its first appearance.
4. Ln.37, why “meanwhile”?
5. Ln.39, what is the evidence showing that FXR activity is “inhibited” instead of “attenuated”?
6. Ln.46, “diabetes in 2021” is what happened in the past. Why “will have”?
7. Ln.47, “will develop” > “are expected to develop”.
8. Ln.53-54, The description “One in 4 patients with type 2 diabetes….” is the estimation based on the past data.
9. Ln.71, eliminate “proved to be”. Bile acids are commonly recognized as one of important signaling molecules.
10. Ln.82, what is “single bile acids”? Why is it plural?
11. Ln.99, “bile acid (CA)” is incorrect.
12. Ln113, “This study aims”
13. Ln.260-264, 20 out of 40 mice were on high-fat diet according to Figure 1A. However, it is not clearly stated in the manuscript.
14. Ln.263, “The diabetic model mice that met the glycemic requirements were….”. Where is the data that shows the satisfaction of the glycemic requirements? Are they in Figure 3?
15. Ln.278, the term “glycolipid” is wrongly used. It is a group of carbohydrates (monosaccharides or oligosaccharides) that are covalently attached to a lipid.
16. Ln.297-299, even the average plasma glucose level of control mice is unusually high. The average plasma glucose level of starved C57BL/6 mice is 60-133 mg/dL (i.e. 3.3-7.4 mmol/L). How did authors collect blood for the measurement of plasma glucose concentration?
17. Ln.307-317, authors stated that T2DM+CUMS mice carried shrunken and deformed islets, compared with those of T2DM mice. However, no significant change was observed in fasting plasma insulin levels between them. How do you explain this inconsistency?
18. Ln.321-322, quantification of expression levels of inflammatory genes in the liver would improve the data.
19. Ln.334-335, authors observed increased lipid accumulation in T2DM+CUMS livers. It appears that lipid accumulation in T2DM+CUMS liver is strikingly increased in Figure 4E. However, no significant differences were indicated in the figure. Is it correct?
20. Ln.343, quantification of Cyp7a1 mRNA level would strengthen the data about the reduction of Fgf15 mRNA levels.
21. In Discussion, authors should discuss about plausible molecular mechanisms of how Fgf15 mRNA levels in the T2DM+CUMS ilea were reduced. Is glucocorticoid involved?

Experimental design

no comment

Validity of the findings

no comment

Additional comments

The manuscript should receive a proofreading by molecular biology/biochemistry experts who speak English as a primary language before resubmission.

Reviewer 3 ·

Basic reporting

no comment

Experimental design

no comment

Validity of the findings

no comment

Additional comments

The study from Cai and colleagues investigated role of bile acids-FXR-FGF15 in diabetic mice suffered with chronic stress. The study is interesting and well-designed. Findings in this study is help to understand role of lipid metabolism in depression in diabetic population. Suggestions are as follows:
1. Metabolic profiling of the brain under T2DM-CUMS is better to be studied.
2. Most of the present findings are phenomena, further studies including in vitro validation and FXR knock-down is suggested.
3. A graphic abstract is suggested.
4. A positive control that demonstrates successful construction of T2DM-CUMS model is better.

---

## Round 0.2 · Minor Revisions

The authors should improve their text conceptually in the following points:
- the first sentence of the abstract should be smoother. “In contrast” should be changed with “However”.

- lines 75-77: streptozotocin treatment has been already reported (lines 71-72). Instead, CUMS protocol should be at least mentioned (modified from Nollet, 2021).

-lines 100-103: the two sentences are repetitive and should be merged into one sentence only.

HDL and LDL should be HDL cholesterol (HDL-C) and LDL cholesterol (LDL-C) in all the context of the whole paper.

English should be thoroughly re-checked by a proficient/professional English speaker. Without this, the manuscript is not publishable, as there are several grammar, typos, and punctuation errors. This issue had been already raised by a previous reviewer and should be seriously considered.

**Language Note:** The Academic Editor has identified that the English language must be improved. PeerJ can provide language editing services - please contact us at copyediting@peerj.com for pricing (be sure to provide your manuscript number and title). Alternatively, you should make your own arrangements to improve the language quality and provide details in your response letter. – PeerJ Staff

Reviewer 3 ·

Basic reporting

no comment

Experimental design

no comment

Validity of the findings

no comment

Additional comments

no comment

---

## Round 0.3 · accepted · Accept

The authors have satisfactorily addressed the last points raised, so that the manuscript is publishable in the present form.